# Evaluation of Adjuvant Systems in Non-Surgical Peri-Implant Treatment: A Literature Review

**DOI:** 10.3390/healthcare10050886

**Published:** 2022-05-11

**Authors:** Andrea Butera, Carolina Maiorani, Simone Gallo, Maurizio Pascadopoli, Adith Venugopal, Anand Marya, Andrea Scribante

**Affiliations:** 1Unit of Dental Hygiene, Section of Dentistry, Department of Clinical, Surgical, Diagnostic and Pediatric Sciences, University of Pavia, 27100 Pavia, Italy; 2Unit of Orthodontics and Pediatric Dentistry, Section of Dentistry, Department of Clinical, Surgical, Diagnostic and Pediatric Sciences, University of Pavia, 27100 Pavia, Italy; simone.gallo02@universitadipavia.it (S.G.); andrea.scribante@unipv.it (A.S.); 3Department of Orthodontics, Saveetha Institute of Medical and Technical Science, Saveetha Dental College, Saveetha University, Chennai 600077, India; avenugopal@puthisastra.edu.kh; 4Department of Orthodontics, Faculty of Dentistry, University of Puthisastra, Phnom Penh 12211, Cambodia; amarya@puthisastra.edu.kh; 5Center for Transdisciplinary Research, Saveetha Institute of Medical and Technical Science, Saveetha Dental College, Saveetha University, Chennai 600077, India

**Keywords:** chlorhexidine, dentistry, erythritol, glycine, laser therapy, mucositis, ozone therapy, peri-implantitis, probiotics

## Abstract

Can the use of lasers, ozone, probiotics, glycine and/or erythritol, and chlorhexidine in combination with non-surgical peri-implant treatment have additional beneficial effects on the clinical parameters? **Objectives**: The non-surgical treatment of peri-implant pathologies is based on mechanical debridement to eliminate bacterial biofilm and reduce tissue inflammation; some additional therapies have been studied to achieve more detailed clinical results. **Materials and methods**: A literature search for publications until January 2022 was conducted. The research question is formulated following the Problem, Intervention, Comparison/Control, and Outcome. Studies investigating adjunctive therapies were included. **Results:** In total, 29 articles were included. Most of the studies did not show any additional benefit of these therapies in the evaluation of bleeding on probing, probing pocket depth, or plaque index; among the proposed treatments, the use of laser was the one most studied in the literature, with the achievement of a reduction of bleeding and pocket depth. More studies would be needed to assess the benefit of other therapies. **Conclusions:** This review showed no significant improvements in the state of health in support of mechanical debridement therapy. However, the few benefits found would deserve to be considered in new clinical studies.

## 1. Introduction

Peri-implant mucositis occurs in approximately 80% of subjects and 50% of the implants, while peri-implantitis in 28–56% of subjects and 12–43% of implant sites [1].

According to these data reported in the literature, peri- implant mucositis affects 43% of patients, variable in a range from 19 to 65%. In comparison, peri-implantitis affects 20% of patients, with a range from 1 to 47%: it should be noted that the data cannot be entirely reliable, as the studies analyzed have used different criteria for the definition of peri-implant pathologies [2,3].

In the new classification of periodontal and peri-implant diseases, conditions are classified for the first time. The definition of implant health emerges, i.e., the absence of clinical signs of inflammation, including bleeding (BoP) and suppuration, the stability of the probe depth (no increase), and the absence of loss beyond the crestal changes in bone level.

Upon taking over one of the following conditions, you can start talking about mucositis and peri-implantitis [4].

Peri-implant mucositis is an inflammatory lesion of the soft tissues surrounding the implant without loss of support bone; the lesion is found laterally to the junctional epithelium but does not extend to the supra crestal connective tissue.

Peri-implantitis, on the other hand, is a pathological condition associated with the presence of bacterial biofilm and characterized by inflammation of the peri-implant mucosa and loss of support bone; in this case, the lesion extends apically to the junction epithelium; there is a greater probing depth (it is good to remember that there is no probing depth that can indicate a health condition of the implant) and a loss of bone assessed radiographically [4,5].

As for the etiology, there are several risk factors involved in the development of peri-implant diseases, such as smoking, periodontal disease, oral hygiene, systemic diseases (diabetes, cardiovascular disease, immunodepression), soft tissue defects, history of one or more failed implants; smoking would seem to be among the leading causes, increasing bone loss up to 0.16 mm/year, as well as diabetes, which, if not controlled, would favor the risk of peri-implantitis development. Other factors may favor the onset of pathologies affecting the implants, such as the positioning of the implants themselves, which may disfavor the proper removal of plaque; it would also seem that the anterior positioning of the implants favors the development of peri-implant pathology: when it is decided to position the implant in this way, it is good to carry out a correct anamnesis, which highlights the history of inflammatory and traumatic pathology and, following dental extraction, to spatiotemporal changes in soft and hard tissues. Considering the factors predisposing to the development of a pathological condition at the implants, it is good to maintain a correct level of oral hygiene at home and implement therapeutic protocols to preserve or restore the state of health [6,7,8,9,10,11].

Among the main treatments proposed in the presence of mucositis and peri-implantitis, aimed at the decontamination of the peri-implant depths and the reduction of bacterial colonization, we indeed find the debridement of the surfaces with ultrasound and specific inserts (peek, carbon, titanium), curettes (Teflon, carbon, titanium) and air-polishing devices. Other supportive therapies, such as lasers and ozone, probiotics, and glycine and erythritol airflow powders, were also proposed [12,13,14,15,16,17]. This review aims to analyze these adjuvant therapies to mechanical debridement in terms of probing depth, bleeding on probing, and plaque index.

## 2. Material and Methods

### 2.1. Focused Question

Can the use of lasers, ozone, probiotics, glycine, erythritol, or chlorhexidine, combined with non-surgical peri-implant treatment, have additional beneficial effects on the clinical parameters?

### 2.2. Eligibility Criteria

*Type of studies*. Randomized controlled clinical trials, prospective clinical trials, and in vivo retrospective clinical trials were included.

*Types of participants*. Participants with the peri-implant disease were considered.

*Type of interventions.* The experimental group was assisted by one or more laser treatments such as diode lasers, Er: YAG laser, Nd: YAG laser, Er, Cr: YSGG laser, LLLT (Low-Level Laser Therapy), PDT (Photodynamic therapy); ozone treatments such as ozone gas, ozone water, ozone gel; treatments with probiotics such as Lactobacillus or Bifidobacterium; treatments with glycine and erythritol air-polishing or perio-polishing; chlorhexidine treatments such as chlorhexidine mouthwash or gel. One or more control groups were administered a placebo or control treatment other than the experimental one.

*Outcome type.* PI (Plaque Index), BoP (bleeding on Probing), and PPD (Probing Pocket Depth) were evaluated.

### 2.3. Search Strategy

The review is based on the research of studies about the PICO model (Population/Problem, Intervention, Comparison/Control, Outcome), detected through bibliographic analysis in electronic databases on Pubmed (MEDLINE) and Google Scholar. Initially, all abstracts of clinical studies were taken into account, which assessed the possible benefit of the addition of laser therapy, ozone therapy, probiotics, glycine and erythritol, and chlorhexidine to non-surgical peri-implant treatment in the treatment of peri-implant diseases.

### 2.4. Research

We performed the search using: “peri-implant diseases”, “peri-implant mucositis”, “peri-implantitis”, “non-surgical peri-implant treatment”, “laser”, “laser AND peri-implant diseases”, “laser AND peri-implant mucositis”, “laser AND peri-implantitis”, “laser AND non-surgical peri-implant treatment”, “ozone”, “ozone AND peri-implant diseases”, “ozone AND peri-implant mucositis”, “ozone AND peri-implantitis”, “ozone AND non-surgical peri-implant treatment”, “probiotics”, “probiotics AND peri-implant diseases”, “probiotics AND peri-implant mucositis”, “probiotics AND peri-implantitis”, “probiotics AND non-surgical peri-implant treatment”, “glycine AND peri-implant diseases”, “glycine AND peri-implant mucositis”, “glycine AND peri-implant diseases”, “glycine AND non-surgical peri-implant treatment”, “erythritol”, “erythritol AND peri-implant diseases”, “erythritol AND peri-implant mucositis”, “erythritol AND peri-implantitis”, “erythritol AND non-surgical peri-implant treatment”, “chlorhexidine”, “chlorhexidine AND peri-implant diseases”, “chlorhexidine AND peri-implant mucositis”, “chlorhexidine AND peri-implantitis”, “chlorhexidine AND non-surgical peri-implant treatment”. We have included patients with peri-implant mucositis or peri-implantitis based on the classification of periodontal diseases proposed by Armitage in 1999 and the new classification presented on 22 June 2018 on the occasion of the Europerio9.

### 2.5. Search Outcome and Evaluation

The first research outcomes were PI, BoP, and PPD [18,19,20]. Information was extracted from each study on (I) participants’ characteristics (age and peri-implant disease); (II) intervention placebo or no treatment or comparison treatment (different from the one tested); (III) outcome (possible benefits of adjunctive treatments); (IV) clinical data examined (PPD, BoP, and PI); (V) follow-up. 

Initially, all abstracts about the topic under review were collected, and then, following a complete reading of the articles, all those not in agreement with the eligibility criteria were discarded. We included only studies in agreement with the criteria of inclusion: (I) studies where the authors did not evaluated at least one of the outcomes taken into account by us, (II) studies where one of the adjunctive treatments taken into account by us were not evaluated as a test group, and (III) in vitro or animal studies were excluded. As a result, articles that did not consider at least one of the selected additional systems and that did not analyze at least one of the clinical incidences (PI, BoP, and PPD) were eliminated.

## 3. Results

A total of 29 studies were therefore identified: 13 articles where the experimental group was treated with laser, 2 articles where the experimental group was treated with ozone, 4 articles where the experimental group was treated with glycine and/or erythritol, 6 articles where the experimental group was treated with probiotics, and 4 articles where the experimental group was treated with chlorhexidine (Figure 1). 

### 3.1. Laser

The thirteen studies [12,21,22,23,24,25,26,27,28,29,30,31,32] selected for the revie are published in English and conducted in many countries; 46.2% of the studies were conducted in Italy [12,26,29] and Germany [22,24,28], 30.8% in Switzerland [21,23] and Sweden [25,31], and 22.1% in Turkey [27], China [30] and Spain [32]. A total of 755 patients were analyzed (an average of 58–59 patients), where 30.8% of the studies involved patients with mucositis [12,26,29,32], 61.5% [21,22,23,25,27,28,30,31] involved patients with peri-implantitis, and only one study considered both issues [24].

Patients were followed in a range of 3–12 months (mostly 6 months of follow-up) of laser treatment: most studies analyzed the beneficial effects of diode lasers [12,26,27,29,32] and Er: YAG lasers [22,24,25,28,31], while 23.1% analyzed the possible efficacy of PDT [21,23,30]; as regards clinical indices, PPD was taken into account in all studies, BoP in 92.3% of studies and PI in 61.5% of studies. Comparing treatment groups, both experimental and control groups showed positive changes during follow-up, showing no statistically significant differences: however, in some studies, major improvements for PPD were found (38.6%) [22,23,24,29,30] and BOP (38.6%) [12,22,28,29,32] in favor of laser treatment groups. 

The results are shown in Table 1.

### 3.2. Ozone

The two studies [13,33] selected for the review are published in English and conducted in Italy and in the USA; a total of 46 patients with mucositis were analyzed.

Patients were followed for 21 days in one of the two studies considered and for 2 months in the other, analyzing ozone gas [13] and ozone water [33], respectively, as regards their possible efficacy in some clinical indices: the first study analyzed PI, noting statistically significant differences between the treatment groups, while the second recounted improvements in terms of PPD, BoP, and PI always in the test group.

The results are shown in Table 2.

### 3.3. Glycine/Erythritol

The four studies [14,34,35,36] selected for the review are published in English and conducted in Holland [14], China [34], Germany [35], and Sweden [36]; a total of 167 patients were analyzed (an average of 41–42 patients), where 50% of the studies involved patients with mucositis [34,36] and 50% involved patients with peri-implantitis [14,35].

Patients were followed in a range of 3–12 months of glycine/erythritol treatment: almost all studies analyzed the beneficial effects of glycine [34,35,36], while only one study evaluated the efficacy of erythritol [14]. PPD was taken into account in all studies, BoP in 75% of studies, and PI in 25% of studies. Comparing treatment groups, both experimental and control groups showed positive changes during follow-up, showing no statistically significant differences. However, in one study, a greater improvement was found in bleeding on probing [35].

The results are shown in Table 3.

### 3.4. Probiotics

The six studies [14,37,38,39,40,41] selected for the review are published in English and conducted in Holland [14], Belgium [37], Japan [38], Italy [39], and Spain [40,41]; a total of 212 patients were analyzed (an average of 35 patients), where 50% of studies involved patients with mucositis [14,39,41], 33.3% involved patients with peri-implantitis [37,38] and only one study considered both problematics [40].

Patients were followed in a range from 6 weeks to 3 months of treatment with probiotics: almost all studies analyzed the beneficial effects of *Lactobacillus reuteri* [14,37,38,40,41], whereas only one study evaluated the efficacy of *Lactobacillus brevis* in combination with *Lactobacillus plantarum* [39]; as regards clinical indices, PPD was taken into account in 66.7% of the studies, BoP in 100% of the studies and PI in 66.7% of the studies. Comparing treatment groups, both experimental and control groups showed positive changes during follow-up, showing no statistically significant differences: however, in one study, an improvement of probing depth was greater [38], while another study showed an improvement in the plaque index only in the experimental group treated with *Lactobacillus reuteri* [37].

The results are shown in Table 4.

### 3.5. Chlorhexidine

The four studies [15,42,43,44] selected for the review are published in English and conducted in Switzerland [15], Brazil [42], Sweden [43], and Spain [44]; a total of 158 patients were analyzed (an average of 39–40 patients) where all studies involved patients with mucositis. Patients were followed in a range of 3–12 months of treatment: the studies examined analyzed the beneficial effects of chlorhexidine in gel or mouthwash at different percentages; with regard to clinical indices, PPD and BoP were taken into account in all studies, and PI in half of the studies involved. By comparing treatment groups, both experimental and control groups showed positive changes during follow-up, showing no statistically significant differences. However, some studies have seen more improvement in PPD [43] and BoP [43,44]. 

The results are shown in Table 5.

### 3.6. Risk of Bias

Randomization, allocation concealment, blinding, outcome data, and outcome recording were evaluated. 

For randomization, participants should be allocated to groups using a true randomization sequence; if studies used the date of birth, admission date, or admission number, it was not evaluated as true randomization.

For allocation concealment, participants and investigators should not be able to predict allocation before the participant is entered into the study, such as centralized allocation (telephone use or web-generated numbering) or sequential numbers contained in anonymous envelopes.

Participants and investigators should be unaware of allocation for blinding to ensure that everyone receives the same amount of care, secondary treatment, or diagnostic testing.

All randomized participants, including those who withdrew from the study or did not receive their intended intervention (intention-to-treat), should be included in the outcome analysis.

For outcome recording, outcomes should be reported for each outcome identified at the outset, primary and secondary; study reports should not focus only on those outcomes that are favorable or those that demonstrate a statistically significant difference between groups.

The risk of bias has been assessed according to the type of randomization and the allocation concealment, the blinding, the outcome data, and the registration of the outcomes, based on the information described in the articles.

For the assessment of risk bias, a color was assigned for each variable analyzed; each color corresponded to a risk value, such as low, moderate, or high. In cases where the information was complete and inherent to the variable considered, a low risk of bias was attributed (green symbol). In cases where the information was scarce or not complete/ missing, a moderate risk was attributed (yellow symbol). Finally, in the cases in which the information was not adequate concerning the variables, for example, randomization based on the date of birth, a high risk of bias was attributed (red symbol).

Table 6, Table 7, Table 8, Table 9 and Table 10 show the risk of bias in the main articles examined; this review presents a relatively low risk of bias. Green symbol: low risk of bias; yellow symbol: moderate risk of bias (also used for lack of information); red symbol: high risk of bias.

## 4. Discussion

Dental implants represent a dental therapy aimed at replacing missing elements in different clinical situations: however, one of the most frequent complications that can lead to the loss of the implant in the presence of peri-implant inflammation, which involves the surrounding hard and soft tissues. This can also be determined by other factors such as the overload of the implant, defects in the materials and techniques used, poor bone quality in the implant area, and systemic pathologies or drug therapies that inhibit bone remodeling [45,46].

The prevention of peri-implant diseases is therefore based on the elaboration of a structured plan that includes individual assessments, minimization of the risk factors, stabilization of the optimal conditions of the surrounding hard and soft tissues, and finally, choice of the correct type of implant, followed by an approach as atraumatic as possible [47]. On the other hand, the treatment includes a non-surgical conservative approach and a surgical approach aimed at decontaminating the implant surfaces. This includes mechanical implant debridement with plastic, titanium, or carbon curettes, ultrasonic instrumentation, or air- and perio-polishing technique [48]. In addition, therapies such as photodynamic or local antiseptic dressings with chlorhexidine, hydrogen peroxide, or sodium percarbonate may support antimicrobial therapy [45,46,47]; this would facilitate the reduction of the pocket depth from 0.5 to 1.0 mm and the bleeding on probing from 15 to 40% [45].

The stabilization of oral hygiene is, therefore, a key element in the prevention of mucositis and peri-implantitis: the goal of therapy should be to resolve the inflammation of the soft tissues and maintain the stability of the supporting bone, trying to instruct the patient in the correct methods of oral hygiene; mechanical debridement and oral hygiene instructions, with or without the addition of supportive therapies, would appear to be as effective as reported into the totality of the studies examined in this review.

The use of laser for the treatment of peri-implant pathologies has been extensively studied, as well as for non-surgical periodontal treatment, leading to a reduction of bleeding on probing from 100 to 43% (following repeated sessions with diode lasers in a 2-year follow-up) [49], but also a reduction in pocket depth (also with photo-modulating therapy) [13]. These results agree with some studies shown in the table (Table 1), which showed improvements in terms of PPD and BoP. Clinical studies conducted by Schwarz, et al., in fact, have reported significant decreases in probing pocket depth (also in pockets of 6 mm, reporting a reduction from 4.6 ± 0.9 mm to 4.1 ± 0.4 mm and from 5.9 ± 0.9 mm to 5.5 ± 0.6 mm) and bleeding on probing, finding improvements even six months after the start of treatment [10,22,24] These results are also found in other studies, such as in a clinical study [25] where the reduction in the probing pocket depth varies from 4.04 ± 0.54 mm at 2.98 ± 0.70 mm and an improvement in the BoP from 44 positive sites to 6 positive sites at the end of the follow-up.

Less significant results compared to the use of the laser as a support to the mechanical debridement in the non-surgical therapy of mucositis and peri-implantitis are reported by studies related to the use of probiotics or gel and mouthwashes based on chlorhexidine. Concerning probiotics, little has been studied in this type of therapy, unlike non-surgical periodontal treatment [50], where improvements in clinical parameters are reported [50,51,52]. Some studies report a significant reduction in probing pocket depth a month after mechanical debridement, with a reduction ranging from 0.5 mm (Table 4) to 1.09 mm [35]; these results are in agreement with another study in the literature, where a postbiotic-based gel appears to be able to reduce inflammatory indices, but they are improvements that would deserve to be evaluated and supported by further research [53]. Less significant in terms of probing pocket depth but also studied as a powerful antibacterial and antimicrobial agent is chlorhexidine, which is often associated with irrigating the peri-implant pockets and would favor the reduction of bleeding probing [44,45]. Despite the improvements observed, both through the use of probiotics (in the analyzed studies *Lactobacilli reuteri* was used) and through the use of chlorhexidine, it would seem that these additional therapies do not lead to more valid results than just mechanical debridement; although several studies in the literature support the benefits of chlorhexidine as an antiseptic agent, it should also be remembered that there are several related adverse events, such as pigmentation of oral soft tissues and teeth, hypersensitivity reactions, taste alteration, burning sensation, ulceration, or erosion of the oral mucosa [54]. The same applies to air- and perio-polishing methods and to the use of ozone, which would not lead to different results compared to the control therapies administered to patients; moreover, as regards the use of ozone, literature is scarce, and consequently, it cannot be said with certainty that it is good therapy in the resolution of implant pathologies or not, although one study has shown improvements in all the clinical parameters analyzed [30].

From the analyzed studies, following the objective of this review, namely, to determine the effectiveness of therapies in terms of bleeding on probing, probing pocket depth, and plaque index, this last parameter does not seem to have been analyzed about particular clinical improvements: only a study with the use of probiotics has favored a modification of the plaque index [36].

Unfortunately, there is a large discrepancy between all the studies that have been analyzed in this review that presents limitations: first of all, it is always difficult to define the probing pocket depth, as there is no standard method used in all studies, and it is a variable also influenced by the thickness of the tissues and the positioning of the implant; from this, it follows that the presence of a pocket is not always synonymous with a disease state of the patient. Another negative point that emerges from the reported articles is that there is no great support literature that can define the effectiveness of these therapies in supporting mechanical debridement. Therefore, it is impossible to perform a valid comparison: also, about the causal therapeutic choice, not all studies were performed in the same way.

What emerges from the clinical results reported by the studies is that these are still valid therapies that have been proven to be good in the treatment of gingivitis and periodontitis. This suggests that further studies should be carried out to validate the findings.

## 5. Conclusions

Based on the results discussed, it can be hypothesized that these additional therapies may provide other clinical benefits in the non-surgical treatment of peri-implant diseases.

Analyzing the effects in terms of improving bleeding on probing, the probing pocket depth, and the plaque index of some therapies supporting the mechanical debridement for the treatment of peri-implant pathologies, although some improvements have emerged, therapies that can provide additional benefits cannot be defined. Considering the reduction in clinical parameters found, these systems should be further studied and analyzed, especially for ozone, glycine/erythritol, probiotics, and chlorhexidine treatment.

## Figures and Tables

**Figure 1 healthcare-10-00886-f001:**
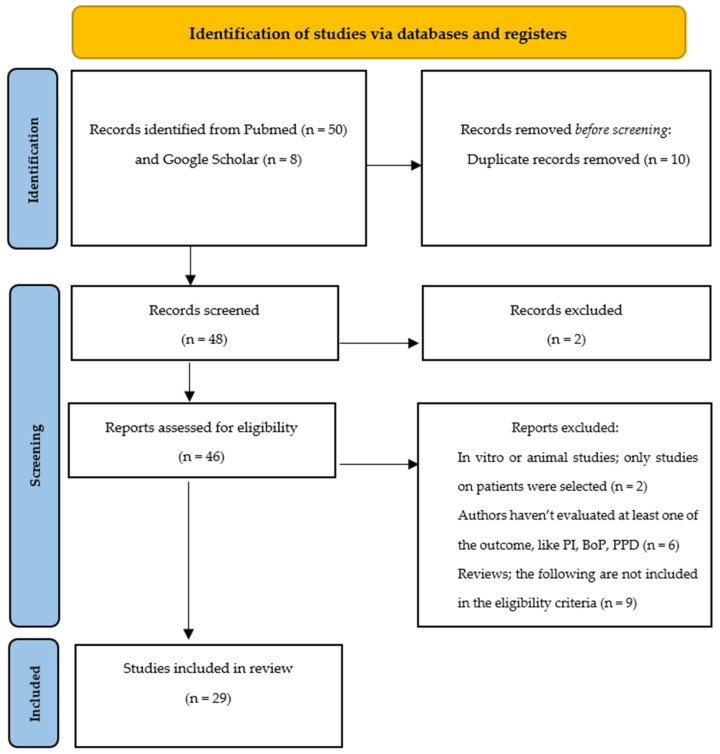
Flow chart of literature research.

**Table 1 healthcare-10-00886-t001:** Laser and non-surgical periodontal therapy for mucositis and peri-implantitis.

Article	Follow-Up	Problem	Intervention	Control	Outcomes
Aimetti et al., 2019 [12]	3 months	Mucositis	Debridement using curettes and ultrasonic devices + diode laser (980 nm, 2.5 W, 10 kHz, 30 s)	Debridement using curettes and ultrasonic devices	Laser was more effective in reducing clinical signs of inflammation
Bassetti et al., 2013 [21]	12 months	Peri-implantitis	Debridement using titanium curettes and glycine air-polishing + PDT (660 nm, 100 mW)	Debridement using titanium curettes and glycine air-polishing + minocycline microspheres	PDT was effective in the reduction of mucosal inflammation from baseline to 6 months and a decrease in PPD from baseline to 9 months
Schwarz et al., 2005 [22]	6 months	Peri-implantitis	Debridement using Er:YAG laser (2.94 mm, 100 mJ/pulse (12.7 J/cm^2^))	Debridement using plastic curettes + chlorhexidine	BOP decreased in the ERL group from 83% at baseline to 31% after 6 months (*p* < 0.001): the difference between the two groups was statistically significant (*p* < 0.001)
Schär et al., 2012 [23]	6 months	Peri-implantitis	Debridement using titanium curettes and glycine air-polishing + PDT (660 nm, 100 mW)	Debridement using titanium curettes and glycine air-polishing + minocycline microspheres	Between-group comparisons revealed no statistically significant differences (*p* > 0.05)
Schwarz et al., 2015 [24]	6 months	Mucositis and peri-implantitis	Debridement using carbon curettes + Er:YAG laser (2.94 μm, 100 mJ/pulse (12.7 J/cm^2^), 10 Hz) (peri-implantitis)	Debridement using carbon curettes + pockets irrigation using 0.1% chlorhexidine solution (mucositis)	Non-surgical treatment of either peri-implant mucositis using MD + CHX or peri-implantitis using ERL at zirconia implants was associated with significant short-term clinical improvements
Persson et al., 2011 [25]	6 months	Peri-implantitis	Debridement using Er:YAG laser (2.94 μm, 100 mJ/pulse (12.7 J/cm^2^))	Debridement using air-abrasive device	The air-abrasive method appeared to have some advantages 1 month after therapy because the countsof pathogens
Mariani et al., 2020 [26]	12 months	Mucositis	Debridement using titanium curettes and power-driven devices + diode laser (980 nm, 2.5 W,10 kHz)	Debridement using titanium curettes and power-driven devices	Diode laser showed little but not statistically significant additional benefits in the treatment of peri-implant mucositis
Arisan et al., 2015 [27]	6 months	Peri-implantitis	Debridement using plastic curettes + diode laser (810 nm (energy density, 3 J/cm^2^; power density, 400 mW/cm^2^; energy, 1.5 J; and spot diameter, 1 mm))	Debridement using plastic curettes	After 6 months, the laser group revealed higher marginal bone loss than the control group. However, in both groups, the microbiota of the implants was found unchanged after 1 month
Schwarz et al., 2006 [28]	12 months	Peri-implantitis	Debridement using Er:YAG laser (2.94 μm, 100 mJ/pulse (12.7 J/cm^2^), 10 Hz)	Debridement using plastic curettes and 0.2% chlorhexidine	Treatment of periimplantitis lesions with laser resulted in a significantly higher BOP reduction than control group
Tenore et al., 2020 [29]	3 months	Mucositis and peri-implantitis	Debridement using titanium curettes and power-driven devices + diode laser (910 nm, 1 W, 50 s)	Debridement using titanium curettes and power-driven devices	The average PPD value for laser group was significantly decreased at 3 months, like BOP
Wang et al., 2019 [30]	6 months	Peri-implantitis	Full-mouth cleansing and glycine powder + PDT (635 nm, 750 mW)	Full-mouth cleansing and glycine powder + 0.9% normal saline	At 1 month, compared with controls, the PD in the PDT group was larger, while at 3 and 6 months, the PDs were smaller (all *p* < 0.001)
Renvert et al., 2010 [31]	6 months	Peri-implantitis	Er:YAG laser (2.94 μm, 100 mJ/pulse [12.7 J/cm^2^], 10 Hz)	Perioflow	A positive treatment outcome, PPD reduction >/−0.5 mm, and gain or no loss of bone was found in 47% and 44% of the perioflow and laser groups, respectively
Sánchez-Martos et al., 2020 [32]	3 months	Mucositis	Debridement with plastic curettes and plastic ultrasound tip + diode laser (810 nm, 1 W, 30 s)	Debridement with plastic curettes and plastic ultrasound tip + sulcus irrigation with 0.12% chlorhexidine and 0.05% cetylpyridinium chloride	The use of diode laser as adjunctive therapy to the conventional treatment of peri-implant mucositis showed promising results, being more effective in reducing the inflammation of the peri-implant tissue

**Table 2 healthcare-10-00886-t002:** Ozone and non-surgical periodontal therapy for mucositis and peri-implantitis.

Article	Follow-Up	Problem	Intervention	Control	Outcomes
McKenna et al., 2013 [13]	21 days	Mucositis	Ozone and saline (1) Ozone and hydrogen peroxide (3)	Hydrogen peroxide and air (2) Air and saline (4)	Significant differences were seen among the treatments (*p* < 0.01) in plaque (F = 16.68), modified gingival (F = 7.86), and bleeding (F = 18.42) indices. O3 + saline and O3 + H2O2 produced optimum gingival health scores
Butera et al., 2021 [33]	2 months	Mucositis	Professional oral hygiene + ozonized water	Professional oral hygiene + pure water	As regards intragroup differences, in Group 1 ozonized water significantly and progressively reduced all the clinical indexes tested, except for PI in the period T1–T2, whereas no significant differences occurred within the control group

**Table 3 healthcare-10-00886-t003:** Air-polishing and non-surgical periodontal therapy for mucositis and peri-implantitis.

Article	Follow-Up	Problem	Intervention	Control	Outcomes
Hantenaar et al., 2021 [14]	12 months	Peri-implantitis	Debridement with erytrhritol	Debridement with ultrasonic device	Three months after therapy, no significant difference in mean BoP (%) between air-polishing and ultrasonic therapy was found
Riben-Grundstrom et al., 2015 [34]	12 months	Mucositis	Debridement with glycine	Debridement with ultrasonic device	At 12 months, there was a statistically significant reduction in mean plaque score, bleeding on probing, and number of periodontal pockets ≥4 mm within the treatment groups compared to baseline
Ji et al., 2014 [35]	3 months	Mucositis	Debridement with ultrasonic scaler (carbon fiber tips) + glycine	Debridement with ultrasonic scaler (carbon fiber tips)	At the 3-month visit, the mean reductions in PD at site level were 0.93 ± 0.93 mm and 0.91 ± 0.98 mm in the test and control groups, respectively (*p* < 0.05), and no significant difference existed between two groups
Sahm et al., 2011 [36]	6 months	Peri-implantitis	Professional oral hygiene + glycine	Debridement using carbon curettes + chlorhexidine	At 6 months, test group revealed significantly higher (*p* < 0.05) changes in mean BOP scores when compared with control treated sites; both groups exhibited comparable PD reductions

**Table 4 healthcare-10-00886-t004:** Probiotics and non-surgical periodontal therapy for mucositis and peri-implantitis.

Article	Follow-Up	Problem	Intervention	Control	Outcomes
Hallström et al., 2015 [15]	6 months	Mucositis	Debridement using titanium curettes + topical application of a droplet of an experimental oil containing *Lactobacillus reuteri* strains DSM 17938 and ATCC PTA 5289 + lozenges containing the same bacteria (one tablet twice a daily)	Debridement using titanium curettes + topical application of a droplet of a placebo oil + placebo lozenges	After 4 and 12 weeks, all clinical parameters were improved in both the test and the placebo group. PPD and BOP were significantly reduced compared with baseline (*p* < 0.05), but no significant differences were displayed between the groups. The clinical improvements persisted 3 months after the intervention.
Laleman et al., 2020 [37]	6 months	Peri-implantitis	Debridement using ultrasound specific tips and titanium curettes + drops containing *Lactobacillus reuteri* DSM 17938 and ATCC PTA 5289	Debridement using ultrasound specific tips and titanium curettes + placebo drops	All clinical parameters were significantly decreased after 12 and 24 weeks. At the implant level, the only statistically significant difference was a greater decrease in plaque levels in the probiotic versus the control group (*p* = 0.002 at 24 wks)
Tada et al., 2017 [38]	6 months	Peri-implantitis	Debridement + azithromycin 500 mg once a day for 3 days + probiotic tablets containing *Lactobacillus reuteri* DSM 17938 and ATCC PTA for 6 months	Debridement + azithromycin 500 mg once a day for 3 days + pacebo tablets for 6 months	PPD in the probiotics group was significantly lower at 4 and 24 weeks than at 0 weeks (*p* < 0.05); a significant decrease did not occur in the placebo group
Mongardini et al., 2016 [39]	6 weeks	Mucositis	Debridement + PDT + probiotic tablets containing *Lactobacillus plantarum* and *Lactobacillus brevis* for 14 days	Debridement + PDT + placebo tablets for 14 days	The combination of plaque removal and PDT, either alone or associated with probiotics, determined a significant reduction of the number of BoP+ sites at 2 and 6 weeks around implants with mucositis
Galofré et al., 2017 [40]	3 months	Mucositis and peri-implantitis	Debridement + probiotic lozenge containing *Lactobacillus reuteri* DSM 17938 and ATCC PTA 5289	Debridement + placebo lozenge	The probiotic *L. reuteri*, together with mechanical therapy, produced an additional improvement over treatment with mechanical therapy alone, both in the general clinical parameters of patients with mucositis and at the level of implants with mucositis or peri-implantitis
Peña et al., 2017 [41]	3 months	Mucositis	Debridement using titanium ultrasound tip + 0.12% chlorhexidine mouthwash + probiotic tablets containing *Lactobacillus reuteri* DSM 17938 and ATCC PTA 5289	Debridement + 0.12% chlorhexidine mouthwash + placebo tablets	Following the administration of probiotics or placebo, the clinical variables, except for probing pocket depth, slightly and progressively increased up to 3 months of follow-up, but without reaching baseline levels

**Table 5 healthcare-10-00886-t005:** Chlorhexidine and non-surgical periodontal therapy for mucositis and peri-implantitis.

Article	Follow-Up	Problem	Intervention	Control	Outcomes
Heitz-Mayfield et al., 2011 [16]	3 months	Mucositis	Debridement using titanium curettes and carbon fiber curettes + 0.5% chlorhexidine gel twice a day for 4 weeks	Debridement using titanium curettes and carbon fiber curettes + 0.5% placebo gel twice daily for 4 weeks	Adjunctive chlorhexidine gel application did not enhance the results compared with mechanical cleansing alone. There was a reduction in PPD and BoP
Menezes et al., 2016 [42]	6 months	Mucositis	Debridement with plastic curettes + 0.12% chlorhexidine solution was used for brushing the dorsum of the tongue for 1 min, rinsing (the last 10 s, the patient should gargle); and subgingival irrigation 3× within 10 min was performed	Debridement with plastic curettes + placebo solution was used for brushing the dorsum of the tongue for 1 min, rinsing (the last 10 s, the patient should gargle); and subgingival irrigation 3× within 10 min was performed	No statistically significant differences were found between the test and control groups at any time
Pulcini et al., 2019 [43]	12 months	Mucositis	Professional prophylaxis + chlorhexidine mouth rinses twice a day	Professional prophylaxis + placebo mouth rinses twice a day	In the test group, there was a 24.49% greater reduction in BOP at the buccal sites than in controls; 58.3% of test implants and 50% of controls showed healthy peri-implant tissues at final visit (*p* > 0.05)
Hallström et al., 2015 [44]	3 months	Mucositis	Debridement using titanium curettes and rubber cup + 0.2% chlorhexidine gel (patients were instructed to brush their teeth once daily with gel)	Debridement using titanium curettes and rubber cup + gel without chlorhexidine (patients were instructed to brush their teeth once daily with gel)	The PPD was significantly reduced (*p* < 0.05) after 12 weeks compared to baseline in the test group but not in the control group

**Table 6 healthcare-10-00886-t006:** Risk of bias of studies on laser.

Article	Adequate Sequence Generated	Allocation Concealment	Blinding	Incomplete Outcome Data	Registration Outcome
Aimetti et al., 2019 [12]					
Bassetti et al., 2013 [21]					
Schwarz et al., 2005 [22]	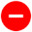				
Schär et al., 2012 [23]					
Schwarz et al., 2015 [24]					
Persson et al., 2011 [25]					
Mariani et al., 2020 [26]					
Arisan et al., 2015 [27]					
Schwarz et al., 2006 [28]					
Tenore et al., 2020 [29]					
Wang et al., 2019 [30]					
Renvert et al., 2010 [31]					
Sánchez-Martos et al., 2020 [32]					

**Table 7 healthcare-10-00886-t007:** Risk of bias of studies on ozone.

Article	Adequate Sequence Generated	Allocation Concealment	Blinding	Incomplete Outcome Data	Registration Outcome
McKenna et al., 2013 [13]					
Butera et al., 2021 [33]					

**Table 8 healthcare-10-00886-t008:** Risk of bias of studies on glycine/erythritol.

Article	Adequate Sequence Generated	Allocation Concealment	Blinding	Incomplete Outcome Data	Registration Outcome
Hantenaar et al., 2021 [14]					
Riben-Grundstrom et al., 2015 [34]					
Ji et al., 2014 [35]	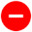				
Sahm et al., 2011 [36]					

**Table 9 healthcare-10-00886-t009:** Risk of bias of studies on probiotics.

Article	Adequate Sequence Generated	Allocation Concealment	Blinding	Incomplete Outcome Data	Registration Outcome
Hallström et al., 2015 [15]					
Laleman et al., 2020 [37]					
Tada et al., 2017 [38]					
Mongardini et al., 2016 [39]					
Galofré et al., 2017 [40]					
Peña et al., 2017 [41]					

**Table 10 healthcare-10-00886-t010:** Risk of bias of studies on chlorhexidine.

Article	Adequate Sequence Generated	Allocation Concealment	Blinding	Incomplete Outcome Data	Registration Outcome
Heitz-Mayfield et al., 2011 [16]					
Menezes et al., 2016 [42]					
Pulcini et al., 2019 [43]					
Hallström et al., 2015 [44]					

## Data Availability

The data presented in this study are available on request from the corresponding author.

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
