# Peer review of "Evaluation of Adjuvant Systems in Non-Surgical Peri-Implant Treatment: A Literature Review"

_healthcare, 2022, doi:10.3390/healthcare10050886_

Round 1
Reviewer 1 Report
The revision of the manuscript was adequate and it is suitable for publication.
Author Response
Thank you very much for your support in improving the quality of the manuscript
Reviewer 2 Report
The manuscript entitled "Evaluation of adjuvant systems in non-surgical peri-implant treatment: a literature review"
review the effect lasers, ozone, probiotics, glycine and/or erythritol, chlorhexidine in combination with non-surgical peri-implant treatment on periodontal parameters.
The paper is interesting; however, some changes should be addressed
Introduction:
The following section should be rewritten: "From a review of the literature published in 2008, we can get an idea of the prevalence of peri-implant disease; from this it emerges that peri-implant mucositis occurred in 40 approximately 80% of the subjects and in 50% of the implants, while peri-implantitis was found in 28% and 56% of subjects and in 12% and 43% of implant sites"
M&M:
The quality assessment should be described deeply in the M&M section to be clearer for the readers. I suggest adding a numerical scale representing the level of bias.
Fig1. The flow chart should be prepared based on PRISMA guidelines; in brackets, you should add the data concerning the reasons pf exclusion/inclusion.
Add laser parameters in table 1. Did Renvert not provide parameters of Er:YAG laser?
Author Response
Please see the attachmen

Reviewer 3 Report
Thank you for the opportunity to review the article “Evaluation of adjuvant systems in non-surgical peri-implant treatment: a literature review.”
The abstract is a bit too long- please shorten it to 200 words as per guidelines.
Please rephrase lines 39-42 to make them sound more scientific.
Lines 43- 45: According to data reported in the literature, other reviews report that mucositis occurs in 43% of the cases, variable in a range ranging from 19 to 65%, while peri-implantitis in around 20% of the cases, with a range ranging from 1 to 47%- please remove the crossed out words
Lines 48-52: please rephrase that part, maybe divide this sentence into two separate ones
Lines 63-68: reference missing to support the statement.
In the section related to Chlorhexidine please refer to the following article: DOI: 10.3390/healthcare10050764.
Generally, the whole manuscript needs a comprehensive language editing as in its present form it is really difficult to read. Some sentences are definitely too long and there is a need for more “scientific” approach to the writing style, I have listen only a few of them. Thera are also some grammatical errors especially with the use of articles and tenses.
Author Response
Please see the attachmen

Round 2
Reviewer 2 Report
The authors respond to all the remarks I stated. I recommend accepting the paper.